# RIM-BP2 regulates Ca²⁺ channel abundance and neurotransmitter release at hippocampal mossy fiber terminals

**Rinako Miyano[1], Hirokazu Sakamoto[2], Kenzo Hirose[2,3], Takeshi Sakaba[1]\***

[1]Graduate School of Brain Science, Doshisha University, Kyoto, Japan; [2]Department of Pharmacology, Graduate School of Medicine, The University of Tokyo, Bunkyo-ku, Japan; [3]International Research Center for Neurointelligence (WPI-IRCN), The University of Tokyo, Bunkyo-ku, Japan

**\*For correspondence:**
tsakaba@mail.doshisha.ac.jp

**Competing interest:** The authors declare that no competing interests exist.

**Abstract** Synaptic vesicles dock and fuse at the presynaptic active zone (AZ), the specialized site for transmitter release. AZ proteins play multiple roles such as recruitment of Ca²⁺ channels as well as synaptic vesicle docking, priming, and fusion. However, the precise role of each AZ protein type remains unknown. In order to dissect the role of RIM-BP2 at mammalian cortical synapses having low release probability, we applied direct electrophysiological recording and super-resolution imaging to hippocampal mossy fiber terminals of RIM-BP2 knockout (KO) mice. By using direct presynaptic recording, we found the reduced Ca²⁺ currents. The measurements of excitatory postsynaptic currents (EPSCs) and presynaptic capacitance suggested that the initial release probability was lowered because of the reduced Ca²⁺ influx and impaired fusion competence in RIM-BP2 KO. Nevertheless, larger Ca²⁺ influx restored release partially. Consistent with presynaptic recording, STED microscopy suggested less abundance of P/Q-type Ca²⁺ channels at AZs deficient in RIM-BP2. Our results suggest that the RIM-BP2 regulates both Ca²⁺ channel abundance and transmitter release at mossy fiber synapses.

## eLife assessment

Miyano et al. study the impact of RIM-BP2 deletion at mossy fiber synapses using direct electro-physiological recordings from mossy terminals and STED super-resolution microscopy. The article addresses an **important** question in the field of synaptic transmission and provides **compelling** evidence demonstrating reduced calcium channel abundance in mossy terminals upon RIM-BP2 removal.

## Introduction

At chemical synapses, an action potential arriving at the presynaptic terminal opens voltage-gated Ca²⁺ channels, and Ca²⁺ influx through voltage-gated Ca²⁺ channels triggers neurotransmitter release from synaptic vesicles within milliseconds. Synaptic vesicle fusion occurs at a specialized region of the presynaptic membrane called the active zone (AZ), where Ca²⁺ channel clusters and vesicle fusion sites are accommodated via AZ scaffold proteins (*Südhof, 2012*). For rapid transmission, accumulation of Ca²⁺ channels, co-localization of Ca²⁺ channels and fusion sites, and molecular priming of synaptic vesicles for fusion are required, and the differences among these factors might provide a basis for synaptic diversity. However, it remains unknown how these factors are regulated by AZ-scaffold proteins.

Rab3-interacting molecule-binding proteins (RIM-BPs) represent one principal, conserved family of AZ proteins and bind to RIMs and Ca²⁺ channels (*Wang et al., 2000*; *Hibino et al., 2002*). RIM-BP

family includes RIM-BP1, RIM-BP2, and RIM-BP3 in mammals, but RIM-BP3 expression is low in the nervous system (*Mittelstaedt and Schoch, 2007*). In *Drosophila* neuromuscular junctions (NMJs), loss of RIM-BPs decreases $Ca^{2+}$ channel density and reduces release probability (*Liu et al., 2011*). Moreover, RIM-BP in *Drosophila* NMJs is necessary for tight coupling of synaptic vesicles to $Ca^{2+}$ channels and replenishment of high release probability vesicles (*Liu et al., 2011*; *Müller et al., 2015*; *Petzoldt et al., 2020*). In mammalian synapses, the observations from KO mice are diverse. In RIM-BP1,2 DKO mice, the coupling between $Ca^{2+}$ channels and synaptic vesicles becomes loose, and action potential-evoked neurotransmitter release is reduced at the calyx of Held synapse (*Acuna et al., 2015*; *Butola et al., 2021*). At hippocampal CA3-CA1 synapses, RIM-BP2 deletion alters $Ca^{2+}$ channel localization at the AZs without altering total $Ca^{2+}$ influx. Besides, RIM-BP1,2 DKO has no additional effect, indicating that RIM-BP2 dominates the function of RIM-BP isoforms (*Grauel et al., 2016*; see also *Krinner et al., 2021*). Tight coupling between $Ca^{2+}$ channels and vesicles is crucial for rapid and efficient transmitter release (*Wadel et al., 2007*).

In contrast, hippocampal mossy fiber-CA3 synapses are characterized by low release probability (*Vyleta and Jonas, 2014*). At hippocampal mossy fiber synapses, gSTED imaging suggests that RIM-BP2 KO alters the Munc13-1 cluster number and distribution. In addition, RIM-BP2 deletion decreases the number of docked vesicles (*Brockmann et al., 2019*). However, because of technical difficulty, direct and quantitative measurements of exocytosis in RIM-BP2 KO terminals have not been performed so far.

To quantify the RIM-BP2 function in mossy fiber-CA3 synapses, we performed whole-cell patch-clamp recordings from WT and RIM-BP2 KO hippocampal mossy fiber boutons. From electrophysiological recordings, we found that the reductions of $Ca^{2+}$ current amplitudes in RIM-BP2 KO terminals. The EPSC and capacitance measurements suggested that the reduction of $Ca^{2+}$ currents was responsible, in part, for that of transmitter release. In addition, they suggested that fusion competence was impaired, which could be overcome by larger $Ca^{2+}$ influx into the terminal. Using STED imaging, we found the abundance of P/Q-type $Ca^{2+}$ channels in terminals to be reduced in RIM-BP2 KO. We suggest that RIM-BP2 regulates the number of P/Q-type $Ca^{2+}$ channels at the AZ and is critical for transmitter release at hippocampal mossy fiber terminals.

## Results

### RIM-BP2 KO reduces presynaptic calcium currents at hippocampal mossy fiber boutons

To investigate the roles of RIM-BP2 in synaptic transmission, we examined the kinetics of exocytosis and $Ca^{2+}$ influx in WT and RIM-BP2 KO synapses using presynaptic whole-cell capacitance measurements (*Figure 1*). When the terminal was depolarized from –80 mV to +10 mV for 10 ms, $Ca^{2+}$ currents and membrane capacitance were recorded as shown in *Figure 1A*. During the depolarizing pulse, capacitance was not measured due to large conductance changes, and the increase was used for measuring synaptic vesicle exocytosis. To determine the size of the RRP and the time course of exocytosis, we measured capacitance changes ($\Delta C_m$) in response to various durations of depolarization. Here, the length of the depolarizing pulse was varied between 5 ms and 100 ms, and $\Delta C_m$ was plotted against pulse duration (*Figure 1B*, *Figure 1—figure supplement 1*, bottom). $\Delta C_m$ became larger as the duration of pulse was prolonged, but the increase started to be saturated at a 100 ms pulse in WT terminals, suggesting depletion of the RRP. $\Delta C_m$ in response to a 100 ms pulse was 52 ± 8.1 fF (n = 9, *Figure 1B*). The amplitude corresponds to the fusion of about 500–600 synaptic vesicles (*Hallermann et al., 2003*; *Midorikawa and Sakaba, 2017*). The time course of $\Delta C_m$ could be fitted by a single exponential with a time constant of 59 ± 12 ms. In RIM-BP2 KO terminals, $\Delta C_m$ in response to a 100 ms pulse was somewhat smaller (41 ± 5.9 fF, n = 8) than that of WT terminals, and the release time constant was somewhat slower (64 ± 14 ms) than in WT terminals. We should note that the time constant is not necessarily reliable because the $\Delta C_m$ values do not reach a plateau value at 100 ms pulse in some terminals. Although the capacitance jumps of KO may be smaller for shorter pulses, capacitance measurements are not sensitive enough to detect small changes (<5 fF, 50 vesicles, see Figures 5 and 6). Nevertheless, the differences in the time course of capacitance were not significantly different between WT and KO (p=0.23 from ANOVA) and the effects on release examined by strong pulses depleting the RRP were relatively minor. The amplitudes of $Ca^{2+}$ currents were smaller in RIM-BP2 KO terminals compared to

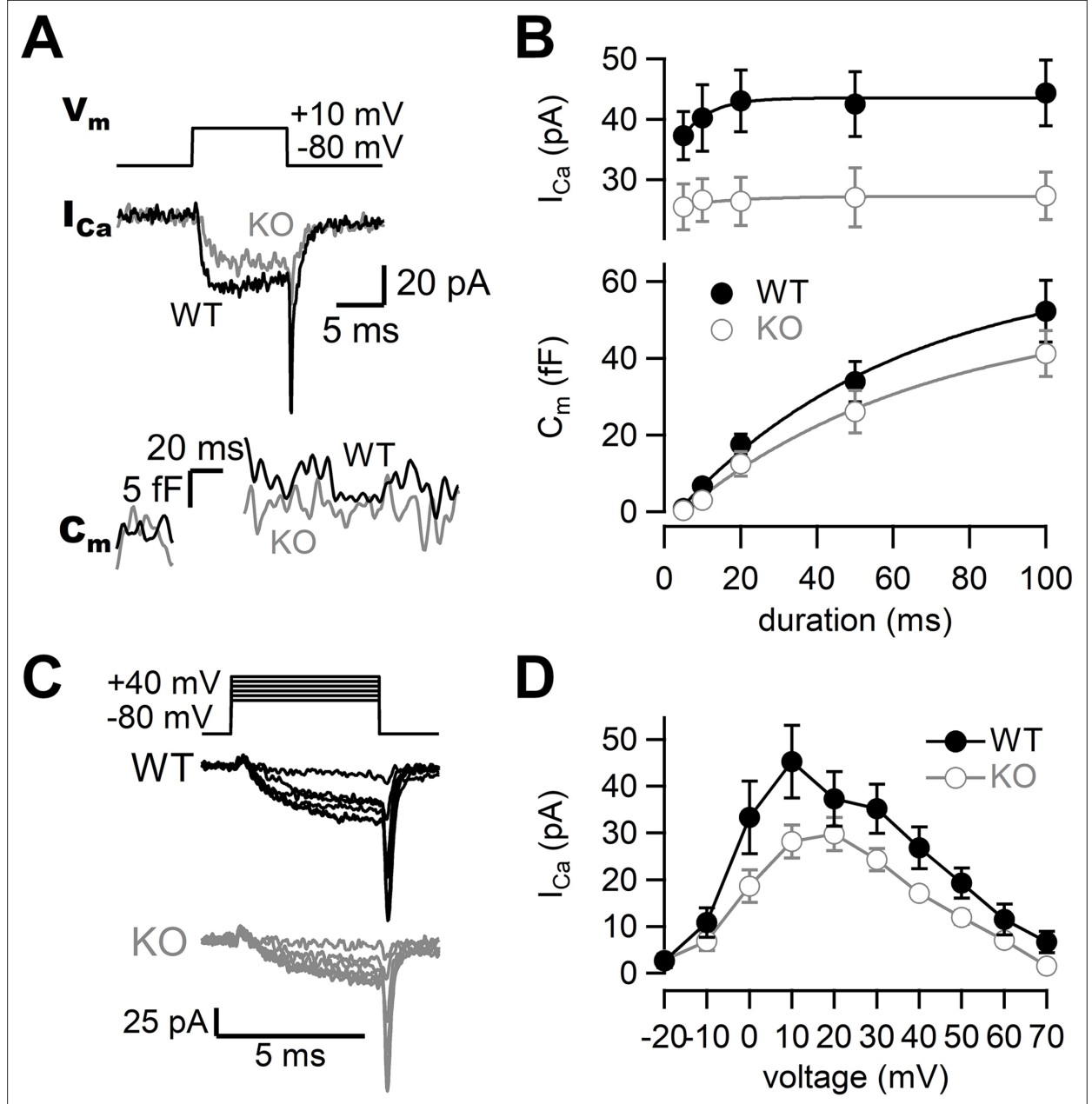

**Figure 1.** Presynaptic calcium currents and synaptic vesicle release in RIM-BP2 KO mice. (**A**) The terminal was depolarized from –80 mV to +10 mV in WT (black) and RIM-BP2 KO (gray) hippocampal mossy fiber boutons. $Ca^{2+}$ currents ($I_{Ca}$) and membrane capacitance ($C_m$) in response to a 10 ms pulse ($V_m$) are shown. (**B**) Top: the peak $Ca^{2+}$ currents are plotted against the pulse duration. WT vs KO, p=0.034 (ANOVA). Bottom: the capacitance jumps are plotted against the pulse duration. WT vs KO, p=0.23 (ANOVA). Black filled circles and gray open circles represent the data from WT (n = 9 from eight animals) and RIM-BP2 KO (n = 8 from six animals), respectively. Each data point represents mean ± SEM. (**C**) Experimental protocol and representative traces for presynaptic $Ca^{2+}$ current measurements in WT (black) and RIM-BP2 KO (gray) boutons. Terminals were sequentially depolarized for 5 ms with 2 s intervals from –80 mV to +70 mV by 10 mV steps. (**D**) Current–voltage relationships of peak $Ca^{2+}$ currents in WT (black filled circle; n = 4–6 from four animals) and RIM-BP2 KO (gray open circle; n = 4–5 from five animals). $I_{Ca}$s were elicited by 5 ms depolarizations. Two curves were significantly different (p<0.01 from mixed model). Each data point represents mean ± SEM. Numerical values of plots are provided in *Figure 1—source data 1*.

The online version of this article includes the following source data and figure supplement(s) for figure 1:

**Source data 1.** Datasets of $Ca^{2+}$ current and capacitance amplitudes presented in *Figure 1* and *Figure 1—figure supplement 1*.

**Figure supplement 1.** Individual value plot of *Figure 1B*.

WT terminals (*Figure 1B*, *Figure 1—figure supplement 1*, top, p<0.02 from ANOVA). The $Ca^{2+}$ current in response to a 100 ms pulse was reduced by ~30% in RIM-BP2 KO terminals (27 ± 3.9 pA, n = 8) compared to WT terminals (44 ± 5.4 pA, n = 9) (p=0.0248, *t*-test, *Figure 1B*). Thus, the results show that RIM-BP2 deletion reduces presynaptic $Ca^{2+}$ influx, but effects on the RRP size and the time course of release measured by capacitance measurements are relatively minor.

To investigate the $Ca^{2+}$ influx in more detail, we examined the voltage dependence of $Ca^{2+}$ currents. *Figure 1C* shows voltage step protocols and representative current traces. In both genotypes, $Ca^{2+}$ currents started to be activated at around –20 mV, and maximal amplitude was observed at ~+10 mV (*Figure 1D*). There was no major difference in the voltage dependence of $Ca^{2+}$ currents between WT and RIM-BP2 KO mice. Consistently, activation time constants of $Ca^{2+}$ currents were similar between WT and RIM-BP2 KO when the terminal was depolarized to +10 mV ($\tau$ = 1.4 ± 0.2 ms in WT, n = 6; $\tau$ = 1.3 ± 0.2 ms in KO, n = 4). These results suggest a reduction in the number of $Ca^{2+}$ channels rather than changes in activation kinetics is responsible for the current decrease observed.

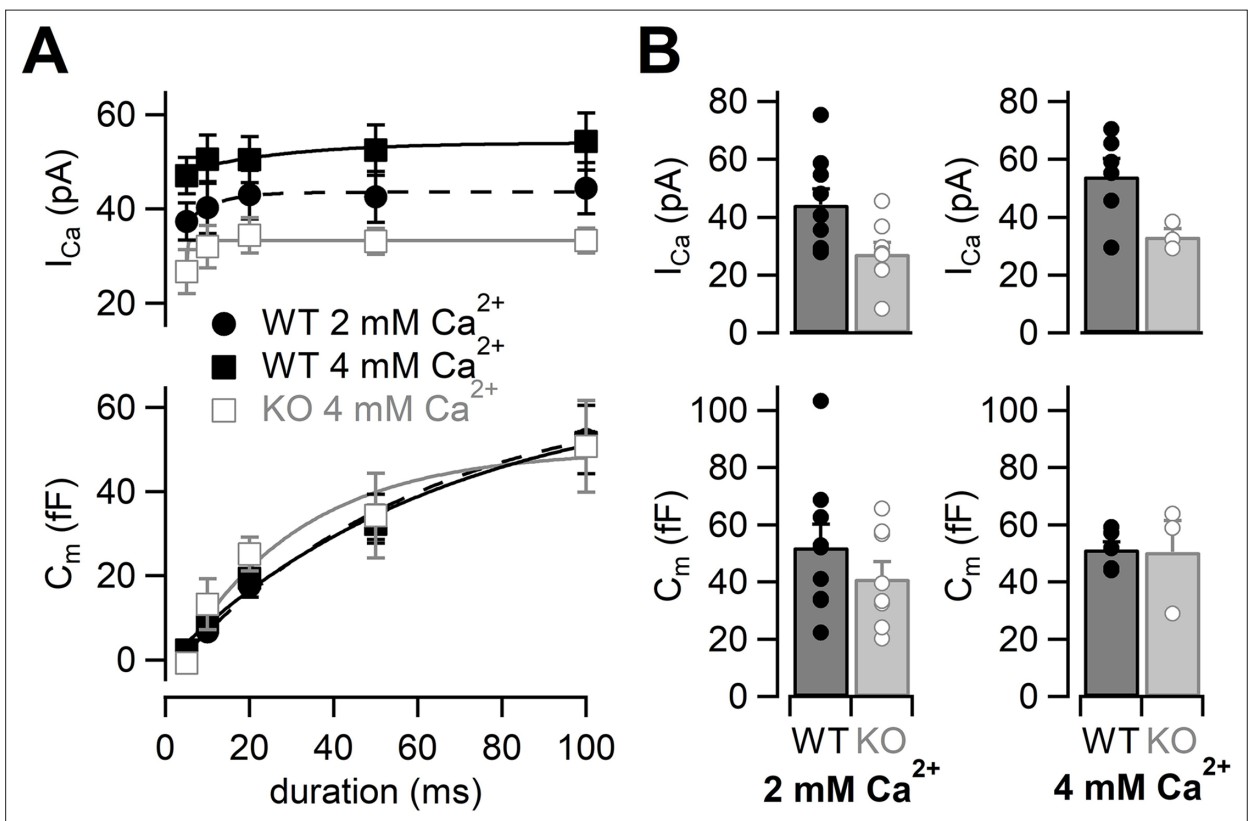

**Figure 2.** The effects of high extracellular calcium concentration on $Ca^{2+}$ currents and capacitance changes. (**A**) The $I_{Ca}$s (top) and the $C_m$s (bottom) are plotted against the pulse duration. Black filled squares and gray open squares represent the data from WT (n = 6–8 from six animals) and RIM-BP2 KO (n = 3–4 from two animals) at 4 mM $[Ca^{2+}]_{ext}$, respectively. For comparison, the WT data in 2 mM $[Ca^{2+}]_{ext}$ are superimposed (black filled circle; n = 9) (the same datasets as *Figure 1B*). Each data point represents mean ± SEM. $I_{Ca}$s were significantly different between WT and KO in 4 mM $[Ca^{2+}]_{ext}$ (p<0.01 from mixed model). The WT $I_{Ca}$s in 2 mM $[Ca^{2+}]_{ext}$ and the KO $I_{Ca}$s in 4 mM $[Ca^{2+}]_{ext}$ were significantly different (p<0.02 from mixed model), but $C_m$ were not significantly different (p=0.57 from mixed model). (**B**) Average $I_{Ca}$s (top) and $C_m$s (bottom) in response to a 100 ms pulse in WT (black bars) and RIM-BP2 KO (gray bars) terminals. Extracellular $Ca^{2+}$ concentration was 2 mM (left) or 4 mM (right). Error bars show SEM. Circles indicate individual values. Numerical values of plots are provided in *Figure 2—source data 1*.

The online version of this article includes the following source data and figure supplement(s) for figure 2:

**Source data 1.** Datasets of $Ca^{2+}$ current and capacitance amplitudes presented in *Figure 2* and *Figure 2—figure supplement 1*.

**Figure supplement 1.** Individual value plot of *Figure 2A*.

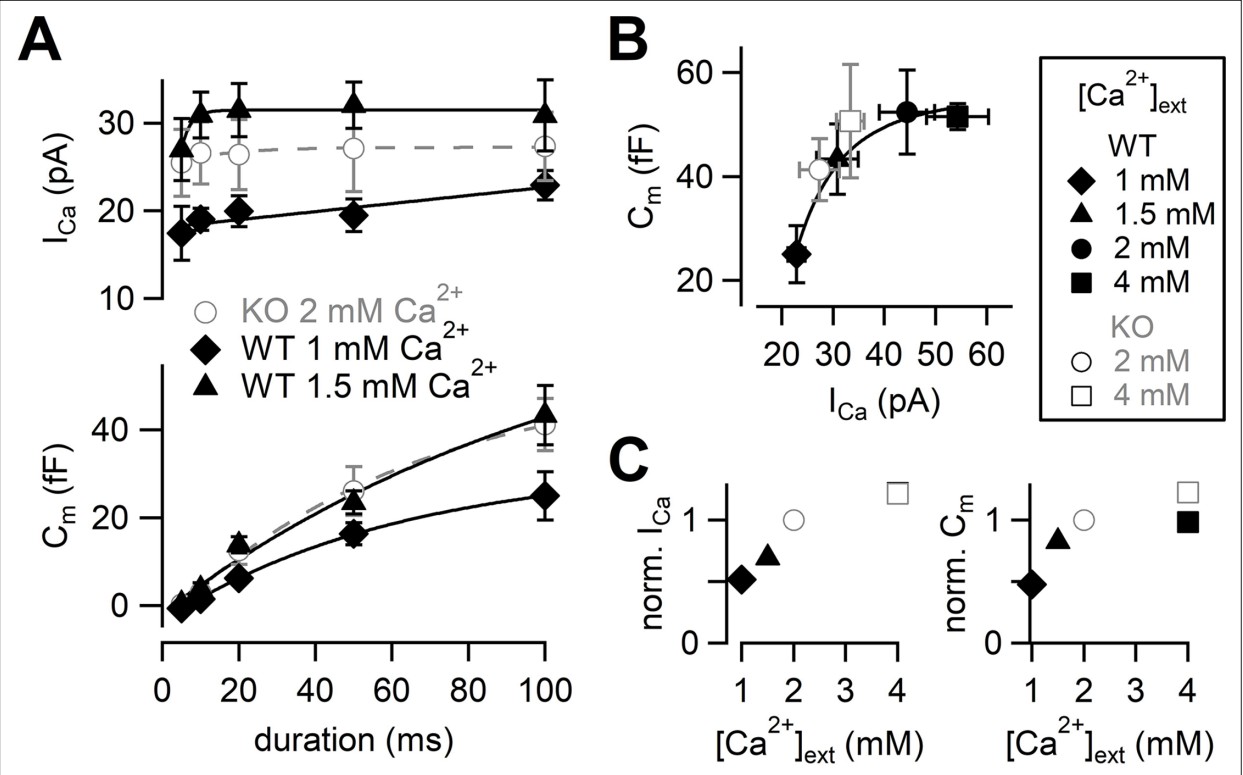

**Figure 3.** Calcium dependence of the release kinetics and the RRP size. (**A**) Top: the relationship between peak $Ca^{2+}$ currents and pulse durations at different $[Ca^{2+}]_{ext}$s. Bottom: the relationship between capacitance jumps and pulse durations at different $[Ca^{2+}]_{ext}$s. Gray open circles, black diamonds, and black triangles represent the data from RIM-BP2 KO at 2 mM $[Ca^{2+}]_{ext}$ (n = 8) (the same datasets as **Figure 1B**), WT at 1 mM $[Ca^{2+}]_{ext}$ (n = 3–5 from four animals) and WT at 1.5 mM $[Ca^{2+}]_{ext}$ (n = 5–7 from six animals), respectively. Each data point represents mean ± SEM. (**B**) Capacitance jumps at various $[Ca^{2+}]_{ext}$s are plotted against calcium current amplitudes. Pulses were 100 ms depolarization from –80 mV to +10 mV. Each data point represents mean ± SEM. Data points were fitted with a Hill equation with n = 3. (**C**) Left: average $I_{Ca}$s at indicated $[Ca^{2+}]_{ext}$s were normalized to the response at 2 mM $[Ca^{2+}]_{ext}$ in each genotype. Right: average $C_m$s at indicated $[Ca^{2+}]_{ext}$s were normalized to the 2 mM $[Ca^{2+}]_{ext}$ response in each genotype. Numerical values of plots are provided in **Figure 3—source data 1**.

The online version of this article includes the following source data and figure supplement(s) for figure 3:

**Figure supplement 1.** Individual value plot of **Figure 3A**.

**Source data 1.** Datasets of numerical values presented in **Figure 3** and **Figure 3—figure supplement 1**.

**Figure supplement 2.** The relationship between synaptic vesicle release and total $Ca^{2+}$ charge.

**Figure supplement 2—source data 1.** Datasets of $Ca^{2+}$ charge and $\Delta C_m$ presented in **Figure 3—figure supplement 2**.

## The effects of extracellular calcium concentration on $Ca^{2+}$ influx and capacitance changes in WT and RIM-BP2 KO

Because $Ca^{2+}$ currents were reduced in RIM-BP2 KO mice, we tested whether the reduction of $Ca^{2+}$ currents could be restored by elevation of the external $Ca^{2+}$ concentration (**Thanawala and Regehr, 2013**). We raised the extracellular $Ca^{2+}$ concentration ($[Ca^{2+}]_{ext}$) from 2 mM to 4 mM (**Figure 2**). $Ca^{2+}$ currents and membrane capacitances were recorded at 4 mM $[Ca^{2+}]_{ext}$, and amplitudes then plotted against pulse duration (**Figure 2A**, **Figure 2—figure supplement 1**). Although $Ca^{2+}$ current elicited by a 100 ms pulse in RIM-BP2 KO was still smaller (33 ± 2.7 pA, n = 3) than that in WT (54 ± 6.1 pA, n = 6) (**Figure 2B**, top) in 4 mM $Ca^{2+}$, $Ca^{2+}$ currents of KO mice at 4 mM $[Ca^{2+}]_{ext}$ were more comparable to those of WT at 2 mM $[Ca^{2+}]_{ext}$.

At 4 mM $[Ca^{2+}]_{ext}$, $\Delta C_m$ in response to a 100 ms pulse in RIM-BP2 KO was 51 ± 11 fF (n = 3) and showed a similar value in WT at 2 mM $[Ca^{2+}]_{ext}$ (**Figure 2B**, bottom). At the same time, $\Delta C_m$ in WT was not altered (52 ± 2.5 fF, n = 6) at higher $[Ca^{2+}]_{ext}$. The time course of exocytosis at 4 mM $[Ca^{2+}]_{ext}$ in KO was similar to that of WT at 2 mM $[Ca^{2+}]_{ext}$ as the time courses were superimposable in **Figure 2A** (p=0.57 from mixed model).

Next, we adjusted Ca$^{2+}$ current amplitudes of WT to that of RIM-BP2 KO by lowering [Ca$^{2+}$]$_{ext}$ (*Figure 3*). The amplitudes of Ca$^{2+}$ currents and capacitance jumps were plotted against pulse duration (*Figure 3A*, *Figure 3—figure supplement 1*). At 1 mM [Ca$^{2+}$]$_{ext}$, Ca$^{2+}$ currents and $\Delta C_m$ were smaller than those of RIM-BP2 KO at 2 mM [Ca$^{2+}$]$_{ext}$. At 1.5 mM [Ca$^{2+}$]$_{ext}$, Ca$^{2+}$ currents became more comparable to those of RIM-BP2 KO at 2 mM [Ca$^{2+}$]$_{ext}$ (p=0.12 from mixed model). Here, the $\Delta C_m$ evoked by a 100 ms pulse (43 ± 6.7 fF, n = 5) was comparable between WT and RIM-BP2 KO (*Figure 3A*). The average time course of capacitance increase of WT at 1.5 mM [Ca$^{2+}$]$_{ext}$ was similar to that of RIM-BP2 KO at 2 mM [Ca$^{2+}$]$_{ext}$ (p=0.88 from mixed model). Therefore, by reducing Ca$^{2+}$ currents, WT data could reproduce similar release time course of RIM-BP2 KO. In *Figure 3B*, $\Delta C_m$ in response to a 100 ms pulse at various [Ca$^{2+}$]$_{ext}$s were plotted against the peak Ca$^{2+}$ current amplitudes. The relationship could be fitted by a Hill plot with n = 3 in WT (*Schneggenburger et al., 1999*). Consistently, when capacitance jumps elicited by various pulse durations are plotted against total Ca$^{2+}$ influx (*Figure 3—figure supplement 2*), the RIM-BP2 KO data were superimposed on the WT data. In *Figure 3C*, relative amplitudes of Ca$^{2+}$ currents and capacitance in response to a 100 ms pulse are plotted. The RIM-BP2 KO data were superimposed on the WT data, suggesting that the RRP size was similar between WT and KO (~50 fF). Taken together, presynaptic capacitance measurements were not able to detect differences in the release rates between WT and KO especially when Ca$^{2+}$ influx in WT and KO was adjusted to be similar. In particular, the RRP sizes measured by the pool depleting stimulation were not altered strongly by RIM-BP2 KO. The results may mean that RIM-BP2 is dispensable for transmitter release or that large Ca$^{2+}$ or large Ca$^{2+}$ influx may overcome the deficit of the release. Alternatively, capacitance measurements may not be sensitive enough to detect the differences between WT and KO at short stimulation, which are relevant for AP-evoked release (see Figures 5 and 6).

## High EGTA experiments suggest unaltered coupling between calcium channels and synaptic vesicles

Previous studies have used Ca$^{2+}$ chelator such as ethylene glycol tetraacetic acid (EGTA) to examine the sensitivity of release to intracellular Ca$^{2+}$ buffers (*Adler et al., 1991*; *Borst and Sakmann, 1996*; *Vyleta and Jonas, 2014*). When the physical coupling between Ca$^{2+}$ channels and synaptic vesicles is tight, EGTA (slow Ca$^{2+}$ chelator) has less effect on release probability, but EGTA is effective when the coupling is loose. Previous experiments have suggested that RIM-BPs regulated the channel–vesicle coupling at the calyx of Held (*Acuna et al., 2015*). If this observation also applies to the mossy fiber synapse, EGTA sensitivity of release should be higher in RIM-BP2 KO terminals. To investigate the coupling at mossy fiber boutons in RIM-BP2 KO, we changed the concentration of EGTA in the patch pipette. In the presence of 5 mM EGTA, we depolarized the terminal from –80 mV to +10 mV for 5 ms and recorded Ca$^{2+}$ currents and membrane capacitance (*Figure 4A*). Ca$^{2+}$ currents and $\Delta C_m$ were plotted against pulse duration (*Figure 4B*, *Figure 4—figure supplement 1*). In both WT and RIM-BP2 KO mice, $\Delta C_m$ were reduced under 5 mM EGTA compared with the control condition of 0.5 mM EGTA. We compared the effect of EGTA on average amplitudes of peak Ca$^{2+}$ currents in response to a 20 ms pulse (*Figure 4D*). Interestingly, Ca$^{2+}$ current amplitudes were ~1.2 times larger at 5 mM EGTA (53 ± 12 pA, n = 5) than at 0.5 mM EGTA (43 ± 5.1 pA, n = 9) in WT, though the data scattered under 5 mM EGTA condition and there was no statistically significant difference (p=0.36, *t*-test). In RIM-BP2 KO, we did not observe such an increase. EGTA might inhibit Ca$^{2+}$ channel inactivation in WT (*von Gersdorff and Matthews, 1996*). In *Figure 4C*, we compared the effect of EGTA on the $\Delta C_m$ in response to a 20 ms pulse. In both genotypes, $\Delta C_m$ values were inhibited by 5 mM EGTA, and the inhibition was somewhat larger in RIM-BP2 KO (EGTA inhibition: p<0.01, genotypes p>0.05 from mixed model).

Ca$^{2+}$ current amplitudes in RIM-B2 KO are smaller than in WT (*Figure 4B*). It is possible that the strong effect of high EGTA on release may be due to reduced Ca$^{2+}$ currents in RIM-BP2 KO. Therefore, we changed [Ca$^{2+}$]$_{ext}$ from 2 mM to 1 mM and adjusted Ca$^{2+}$ current amplitudes of WT to the RIM-BP2 KO level. The capacitance increase and the amplitudes of Ca$^{2+}$ currents became similar to those of RIM-BP2 KO (*Figure 4C and D*). These results indicated that WT and RIM-BP2 KO terminals had similar sensitivity of neurotransmitter release to EGTA. Thus, it is unlikely that changes in the coupling distance are responsible for the reduced exocytosis of the mutants.

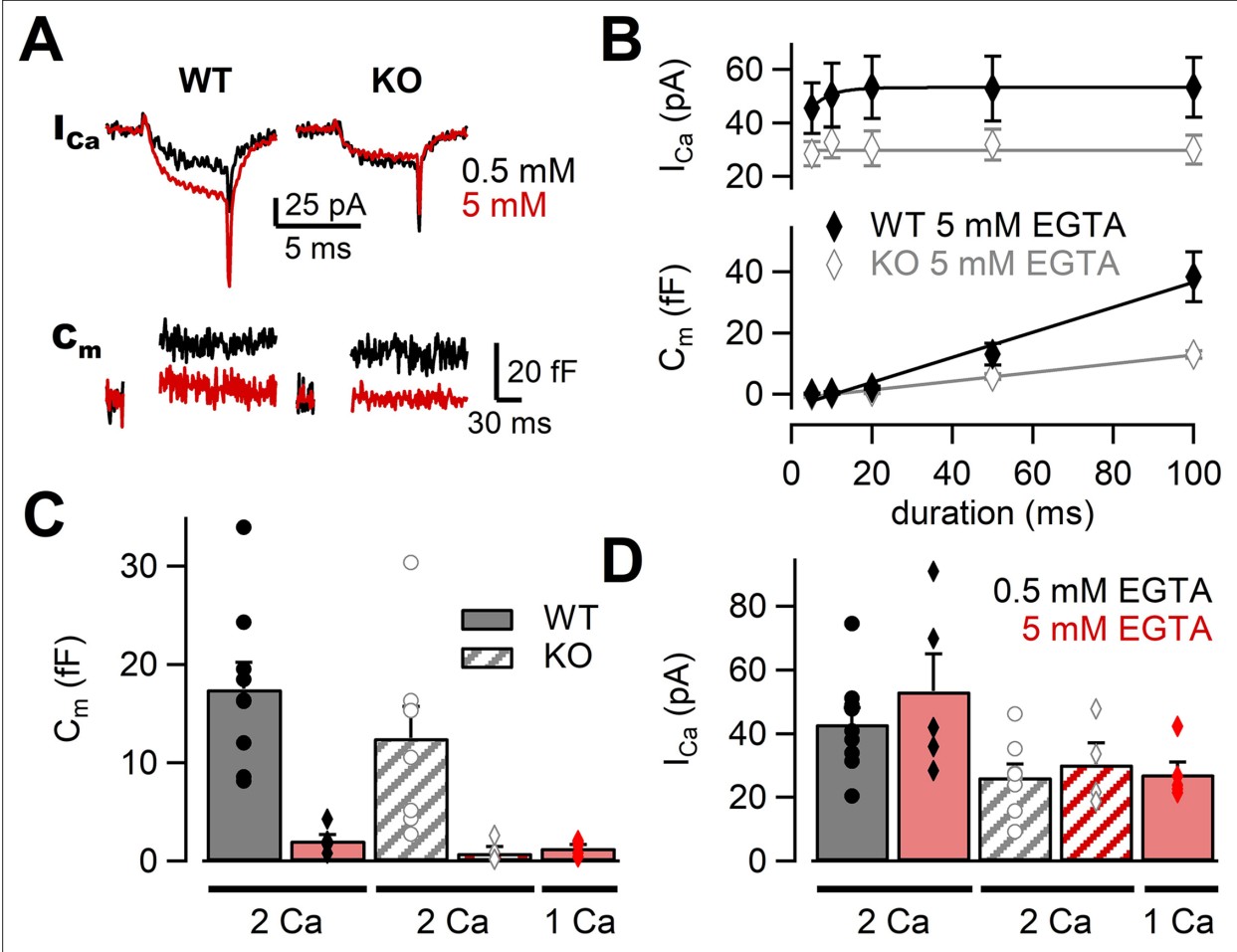

**Figure 4.** The effects of high EGTA on calcium currents and synaptic vesicle exocytosis. (**A**) Example traces in response to a 5 ms depolarizing pulse to +10 mV in WT (left) and RIM-BP2 KO (right) boutons. $Ca^{2+}$ current ($I_{Ca}$) and membrane capacitance ($C_m$) recorded with 0.5 mM EGTA (black) or 5 mM EGTA (red) in the patch pipette are shown. Note that each trace was obtained from different terminals but the traces were superimposed. (**B**) The $I_{Ca}$s (top) and $C_m$s (bottom) are plotted against the pulse duration. The patch pipette contained 5 mM EGTA. Black filled diamonds and gray open diamonds represent the data from WT (n = 4–5 from four animals) and RIM-BP2 KO (n = 4 from three animals), respectively. Each data point indicates mean ± SEM. (**C, D**) Average $C_m$s (**C**) and $I_{Ca}$s (**D**) elicited by a 20 ms pulse in WT (filled bars) and RIM-BP2 KO (hatched bars). The extracellular $Ca^{2+}$ concentration was 1 mM (from five animals) or 2 mM. The intracellular solution contained either 0.5 mM EGTA (black) or 5 mM EGTA (red). Error bars show SEM. Circles and diamonds indicate individual values. Numerical values of plots are provided in *Figure 4—source data 1*.

The online version of this article includes the following source data and figure supplement(s) for figure 4:

**Source data 1.** Datasets of $Ca^{2+}$ current and capacitance amplitudes presented in *Figure 4* and *Figure 4—figure supplement 1*.

**Figure supplement 1.** Individual value plot of *Figure 4B*.

## Smaller AP-evoked EPSCs in KO could not be explained entirely by the reduced $Ca^{2+}$ influx

While $Ca^{2+}$ current amplitudes were clearly reduced in KO, capacitance jumps were reduced little. Capacitance measurements could only detect large amounts of exocytosis (~5 fF, 50 vesicles), and the changes relevant for the AP-evoked release (<10 vesicles per AP per synapse) could not be detected. Moreover, large $Ca^{2+}$ influx used for the stimulation protocol of *Figures 1–3* could overcome the deficits of RIM-BP2 KO. In order to detect the changes related to physiological conditions, we measured the AMPA-EPSCs evoked by electrical mossy fiber stimulation at CA3 pyramidal cells (*Figure 5*). As seen in *Figure 5A and B* , the EPSCs were evoked by dual stimulation of mossy fibers. The mossy fiber-evoked responses were confirmed by the relatively faster EPSC kinetics and sensitivity to DCG-IV (1 μM), an mGluR2 agonist (remaining response of <30%, 0.23 ± 0.01 for WT [n = 5], 0.29 ± 0.02 for KO [n = 5]). Because multiple fibers were stimulated, we could not estimate the number of vesicles

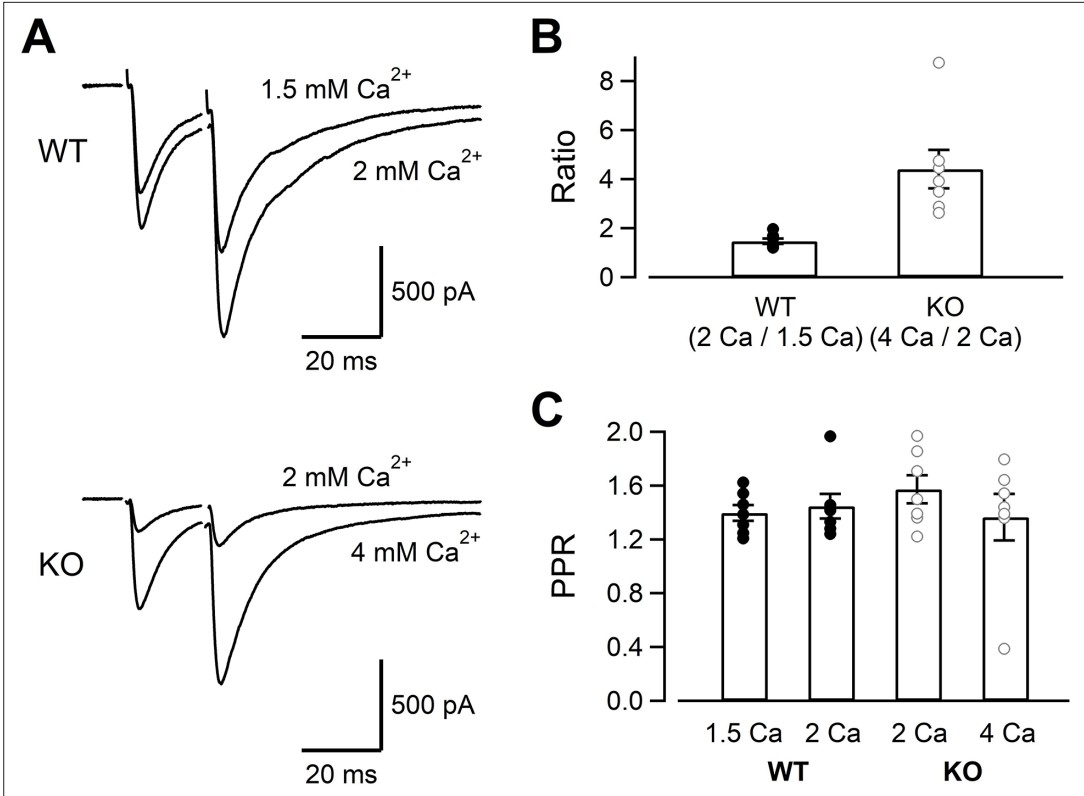

**Figure 5.** The evoked EPSC amplitudes in WT and KO, and their sensitivity to the extracellular Ca²⁺. (**A**) The mossy fiber-evoked EPSCs measured at CA3 pyramidal cells. The fibers were stimulated twice with an interval of 20 ms. The responses of WT in 1.5 mM and 2 mM [Ca²⁺]ₑₓₜ (n = 7 cells from five animals), as well as those of KO in 2 mM and 4 mM [Ca²⁺]ₑₓₜ (n = 7 cells from six animals) are shown. The stimulus artifacts were blanked. (**B**) The amplitude ratios of WT EPSCs (2 mM/1.5 mM [Ca²⁺]ₑₓₜ) and KO EPSCs (4 mM/2 mM [Ca²⁺]ₑₓₜ) are shown. The concentrations were chosen to set the Ca²⁺ influx similar between WT and KO from *Figures 1–3*. (**C**) The paired pulse ratios under four conditions (WT in 1.5 mM and 2 mM [Ca²⁺]ₑₓₜ, KO in 2 mM and 4 mM [Ca²⁺]ₑₓₜ). Numerical values of plots are provided in *Figure 5—source data 1*.

The online version of this article includes the following source data for figure 5:

**Source data 1.** Datasets of numerical values presented in *Figure 5*.

released per one synapse. Nevertheless, when fibers were stimulated similarly, the EPSC amplitudes of KO were much smaller (893 ± 202 pA in WT, n = 7; 189 ± 73 pA in KO, n = 7) (p=0.011). In *Figure 2*, presynaptic recordings indicated that Ca²⁺ current amplitudes of KO at 4 mM [Ca²⁺]ₑₓₜ became similar to those of WT at 2 mM [Ca²⁺]ₑₓₜ (*Figure 2*). The EPSCs became larger when the [Ca²⁺]ₑₓₜ was raised from 2 mM to 4 mM in KO (*Figure 5A*, top, see also *Figure 6—figure supplement 1*) and became more similar to those of WT. In *Figure 3*, presynaptic recording showed that I_Cas of WT at 1.5 mM [Ca²⁺]ₑₓₜ were similar to those of KO at 2 mM [Ca²⁺]ₑₓₜ. However, the WT EPSCs at 1.5 mM [Ca²⁺]ₑₓₜ seemed larger than KO at 2 mM [Ca²⁺]ₑₓₜ (*Figure 5A*, bottom, see also *Figure 6—figure supplement 1*). Changing the [Ca²⁺]ₑₓₜ from 1.5 mM to 2 mM in WT and from 2 mM to 4 mM in KO should have the same potentiation effect on the EPSCs if the reduced Ca²⁺ current amplitudes are the only mechanism for reduced EPSCs in KO. In *Figure 5B*, we calculated two types of potentiation ratios: in WT, the ratio of the EPSC response in 2 mM to that in 1.5 mM [Ca²⁺]ₑₓₜ was calculated in each cell. In KO, the ratio of the EPSC response in 4 mM to that in 2 mM [Ca²⁺]ₑₓₜ was calculated in each cell. When these two values were compared, the ratios were much larger in KO (*Figure 5B*, p<0.01, *t*-test). Pr is more sensitive to the [Ca²⁺]ₑₓₜ in the low end of the dose–response curve because of the third to fourth power dependence of release on [Ca²⁺]ₑₓₜ at the low end (*Dodge and Rahamimoff, 1967*; *Schneggenburger and Neher, 2000*; *Bollmann et al., 2000*). Therefore, the result suggests lower release probability in KO even when the amounts of Ca²⁺ influx were adjusted to be similar. When the paired pulse ratios were compared, all four conditions (WT in 1.5 mM and 2 mM [Ca²⁺]ₑₓₜ, KO in 2 mM and 4 mM [Ca²⁺]ₑₓₜ)

provided similar values (*Figure 5C*). The paired pulse ratio should be inversely correlated to Pr, in principle. However, paired pulse facilitation in mossy fiber synapses could be mediated by multiple effects, including $Ca^{2+}$-independent broadening of AP waveforms (*Geiger and Jonas, 2000*) and $Ca^{2+}$ buffer saturation, which is more pronounced and increases the paired-pulse ratios at higher $[Ca^{2+}]_{ext}$ (*Blatow et al., 2003*). Therefore, the value might not be sensitive to the $[Ca^{2+}]_{ext}$.

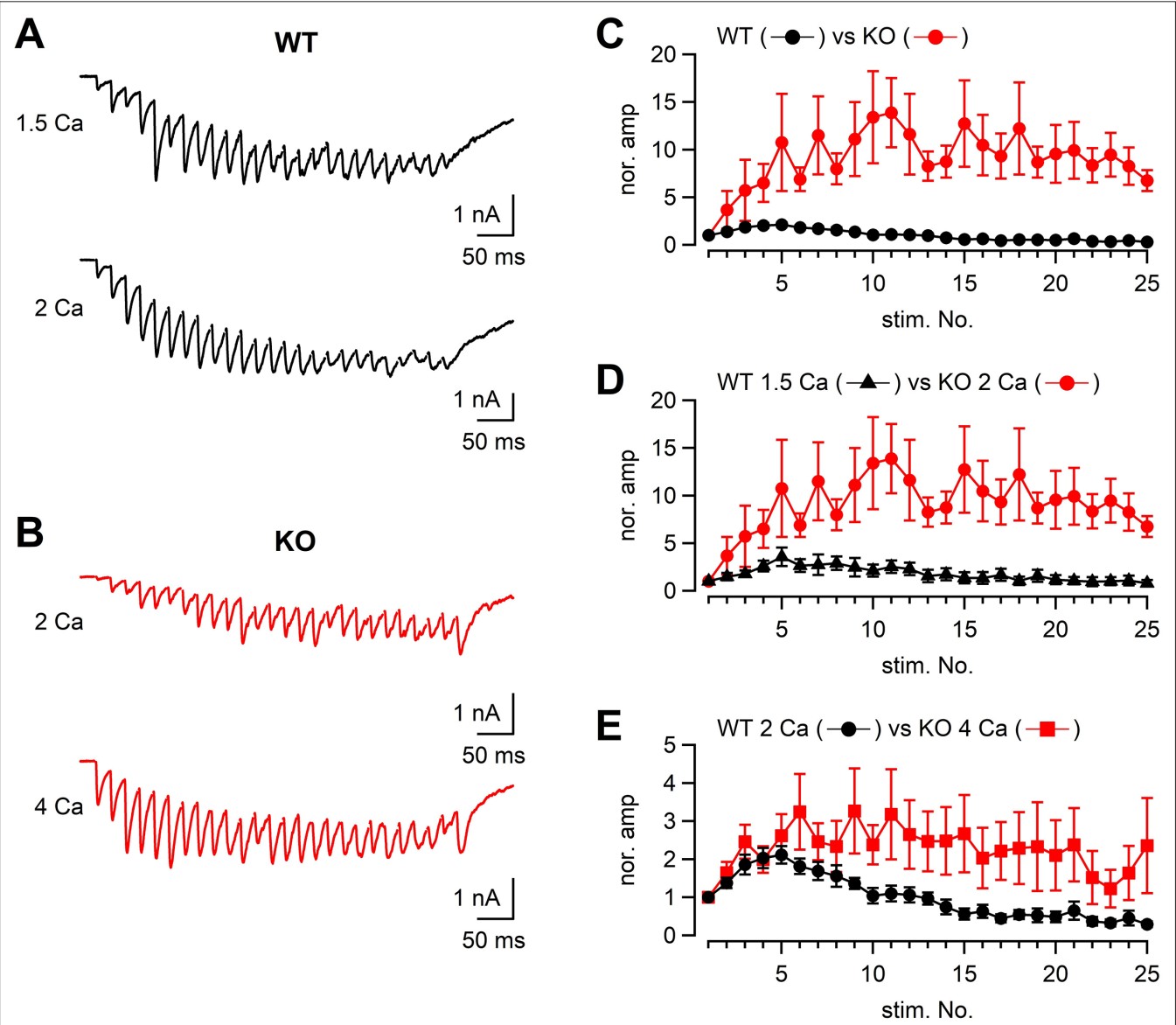

**Figure 6.** The time course of synaptic facilitation/depression in WT and KO. (**A**) Mossy fibers were stimulated at 50 Hz (25 times) and the evoked EPSCs were measured in 1.5 mM (n = 5 cells from four animals) and 2 mM $[Ca^{2+}]_{ext}$ (n = 7 cells from five animals) in WT. The stimulus artifacts were blanked. (**B**) Mossy fibers were stimulated at 50 Hz (26 times in this particular example), and the evoked EPSCs were measured in 2 mM (n = 7 cells from six animals) and 4 mM $[Ca^{2+}]_{ext}$ (n = 8 cells from six animals). (**C**) The time course of EPSC amplitudes during a 50 Hz train. The data were obtained from WT and KO in 2 mM $[Ca^{2+}]_{ext}$. The time courses were significantly different (p<0.01 from ANOVA). (**D**) The same as (**C**), but the data were obtained from WT in 1.5 mM $[Ca^{2+}]_{ext}$ and KO in 2 mM $[Ca^{2+}]_{ext}$. The time courses were significantly different (p<0.05 from ANOVA). (**E**) The same as (**C**), but the data were obtained from WT in 2 mM $[Ca^{2+}]_{ext}$ and KO in 4 mM $[Ca^{2+}]_{ext}$. Numerical values of plots are provided in *Figure 6—source data 1*. The time courses were not significantly different (p=0.12 from ANOVA). Numerical values of plots are provided in *Figure 6—source data 1*.

The online version of this article includes the following source data and figure supplement(s) for figure 6:

**Source data 1.** Datasets of numerical values presented in *Figure 6* and *Figure 6—figure supplements 1 and 2*.

**Figure supplement 1.** The EPSC amplitudes plotted against the stimulus number.

**Figure supplement 2.** The individual data of *Figure 6*.

We also analyzed the EPSCs in response to a stimulus train (50 Hz × 25 or 26 pulses) both in WT and KO (*Figure 6*) under different external $Ca^{2+}$ concentrations. When the WT and KO responses were compared, the time course of facilitation/depression was quite different between WT and KO (*Figure 6A–C*). In 2 mM $[Ca^{2+}]_{ext}$, WT responses showed initial facilitation (approximately twofold) followed by depression, whereas KO responses showed pronounced facilitation of approximately tenfold. The difference could be due to both reduced $Ca^{2+}$ currents and impaired fusion competence in KO.

To adjust presynaptic $Ca^{2+}$ influx to the similar level as the KO terminals, the $[Ca^{2+}]_{ext}$ was reduced to 1.5 mM in WT. Under this condition, presynaptic $Ca^{2+}$ current amplitudes were expected to be similar between WT and KO (in 2 mM $[Ca^{2+}]_{ext}$). If the reduced $Ca^{2+}$ influx is only responsible for smaller EPSCs in KO, we will observe similar time course of train responses between WT and KO. However, facilitation was much less in the WT condition (at 1.5 mM $[Ca^{2+}]_{ext}$, *Figure 6D*, p<0.05 from ANOVA), suggesting the mechanism independent of $Ca^{2+}$ influx. In order to obtain the same $Ca^{2+}$ influx as that of WT in 2 mM $[Ca^{2+}]_{ext}$, the $[Ca^{2+}]_{ext}$ was increased to 4 mM in KO. The time course of facilitation was somewhat larger in the KO condition compared with that WT in 2 mM $[Ca^{2+}]_{ext}$. Because of the data scatter in KO, the time course was statistically not significant (p>0.05 from ANOVA). Because multiple fibers were stimulated, the absolute EPSC amplitudes depend on not only the release probability and the number of release sites per synapse, but also the number of stimulated synapses. Nevertheless, when the absolute amplitudes were plotted against the stimulus number, the differences were most prominent at the first responses in the train and the differences were diminished later in the train, suggesting that the initial release probability was likely to be lower at RIM-BP2 KO synapses, but the release could be restored during repetitive stimulation (*Figure 6—figure supplement 1*). Restoration of release is consistent with minor difference of capacitance changes in response to the depolarization (*Figures 1–4*).

## STED microscopy is consistent with less abundance of P/Q-type $Ca^{2+}$ channels in RIM-BP2 KO

Our electrophysiological data suggest that RIM-BP2 may regulate the abundance of $Ca^{2+}$ channels. In order to directly study the $Ca^{2+}$ channel abundance at the AZ, we performed STED microscopy analysis of $Ca^{2+}$ channels in WT and RIM-BP2 KO hippocampal mossy fiber terminals. We here focused on P/Q-type and N-type $Ca^{2+}$ channels because both $Ca^{2+}$ channel types are relevant for transmitter release at this synapse (*Castillo et al., 1994*; *Pelkey et al., 2006*; *Li et al., 2007*). We performed immunohistochemistry on thin brain cryosections from WT and RIM-BP2 KO mice. STED microscopy confirmed complete loss of RIM-BP2 proteins in KO terminals (*Figure 7—figure supplement 1*). We then immunohistochemically labeled the α-subunit of either P/Q-type $Ca^{2+}$ channel (Cav2.1) or N-type $Ca^{2+}$ channel (Cav2.2) and Munc13-1 (AZ/release site marker, *Böhme et al., 2016*; *Sakamoto et al., 2018*) with VGLUT1 (presynaptic marker) and PSD-95 (postsynaptic marker) (*Figure 7A and B*, *Figure 7—figure supplement 2*). In the stratum lucidum of the hippocampal CA3 region, mossy fibers form large synapses containing multiple and clustered AZs on CA3 pyramidal cells, and also small en passant and filopodial synapses containing a single AZ on interneurons (*Acsády et al., 1998*; *Rollenhagen et al., 2007*). Consistent with this, large VGLUT1-positive terminals rarely contacted interneurons marked with two major interneuron markers, mGluR1α and GluA4 (*Figure 7—figure supplement 3*). Thus, we identified AZs using Munc13-1 STED images and quantified the signal intensity of Cav2.1 or Cav2.2 only at AZs clustered in large VGLUT1-positive terminals in the CA3 stratum lucidum, allowing us to confine the analysis to large mossy fiber terminals innervating onto CA3 pyramidal cells. The total signal intensity of Cav2.1 at the AZ was 22% lower in RIM-BP2 KO mice (n = 4 animals) than in WT mice (n = 4 animals) (p=0.0286; *Figure 7C*, left). These data are consistent with the reduction of $Ca^{2+}$ currents in RIM-BP2 KO mice (*Figure 1*), though STED microscopy is nonlinear and can only provide qualitative differences regarding intensity. In addition, the spatial resolution of STED microscopy (especially z axis) does not assign the signals to the large mossy fiber terminal rigidly. The area, length, and width of Cav2.1 clusters in the AZ were not different between WT and RIM-BP2 KO terminals (*Figure 7—figure supplement 4*). The analysis in this study might not have spatial resolution to detect the differences. The analysis from the confocal microscopy also indicated that the signal intensity of Cav2.1 was reduced in KO, consistent with the STED data (*Figure 7—figure supplement 5*).

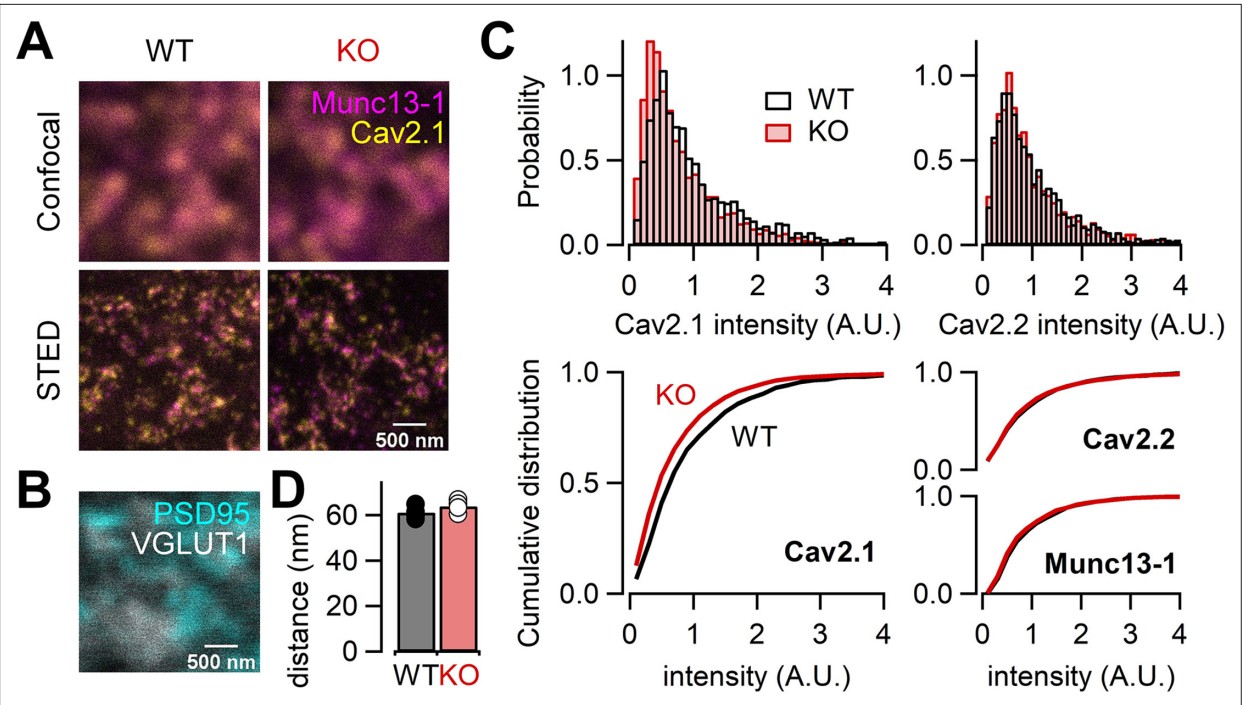

**Figure 7.** RIM-BP2 deletion alters the signal intensity of Cav2.1 clusters within the active zone (AZ). (**A**) Confocal (top) and STED (bottom) images of Munc13-1 (magenta) and Cav2.1 (yellow) clusters at hippocampal mossy fiber terminals of WT (left) and RIM-BP2 KO (right) mice. Scale bar: 500 nm. (**B**) Confocal images of PSD95 (cyan) and VGLUT1 (white) to identify glutamatergic synapses in CA3 stratum lucidum: mossy fiber-CA3 synapses. Scale bar: 500 nm. The image was taken from same region shown in (**A**, left). (**C**) Histograms (top) and cumulative distribution plots (bottom) of the total signal intensity of Cav2.1 (left), Cav2.2, and Munc13-1 (right) at AZs in WT (black) and RIM-BP2 KO (red) mice. (**D**) The average nearest-neighbor distance between Cav2.1 and Munc13-1 clusters in WT (black; n = 4 animals) and RIM-BP2 KO (red; n = 4 animals) mice. Several hundreds of AZs per image were analyzed. Data show the average value of distance per animal, and error bars represent SEM. Each data point indicates individual values. Numerical values of plots are provided in *Figure 7—source data 1*.

The online version of this article includes the following source data, source code, and figure supplement(s) for figure 7:

**Source code 1.** Custom-designed programs in Mathematica.

**Source data 1.** Datasets of numerical values presented in *Figure 7*.

**Figure supplement 1.** STED microscopy confirmed loss of RIM-BP2 proteins in KO terminals.

**Figure supplement 2.** STED imaging of Cav2.2 at hippocampal mossy fiber terminals.

**Figure supplement 3.** Immunohistochemical characterization of large VGLUT1-positive terminals in the CA3 stratum lucidum.

**Figure supplement 4.** The area, length, and width of Cav2.1 cluster.

**Figure supplement 4—source data 1.** Datasets of numerical values presented in *Figure 7—figure supplement 4*.

**Figure supplement 5.** Quantification of the signal intensity using confocal images.

**Figure supplement 5—source data 1.** Datasets of numerical values presented in *Figure 7—figure supplement 5*.

In contrast to Cav2.1, the total signal intensity of Cav2.2 and Munc13-1 did not significantly differ between WT and RIM-BP2 KOs (p=0.3143 and 0.0831) (*Figure 7C*, right), though it remains possible that we failed to detect some changes. Furthermore, we found that the number of Cav2.1 clusters within the AZ identified by STED deconvolution analysis was not altered (2.5 ± 0.11 in WT, n = 4 animals; 2.3 ± 0.02 in KO, n = 4 animals) (p=0.114). These data are consistent with the idea that RIM-BP2 KO reduces the abundance of P/Q-type $Ca^{2+}$ channels per cluster. To optically estimate physical distances between $Ca^{2+}$ channels and release sites, we next analyzed the nearest-neighboring distance between Cav2.1 and Munc13-1 clusters. These distances were unchanged (61 ± 1.4 nm in WT, n = 4 animals; 64 ± 1.3 nm in KO, n = 4 animals) (p=0.200) (*Figure 7D*), consistent with the similar sensitivity of release to EGTA between WT and RIM-BP2 KO mice. However, the resolution of STED microscopy cannot allow detection of small differences of the cluster number and the distance between WT and KO. From these results together with presynaptic recordings, we suggest that decreased $Ca^{2+}$ influx

in RIM-BP2 KO hippocampal mossy fiber terminals is caused by less abundance of P/Q-type $Ca^{2+}$ channels at the AZ.

## Discussion

Accumulation of $Ca^{2+}$ channels at the AZ, and efficient vesicle docking and priming are essential factors for fast synaptic vesicle exocytosis. AZ proteins are critical for fast synchronous release and synaptic diversity. RIM-BPs have been implicated in recruitment of $Ca^{2+}$ channels to the AZ (*Kaeser et al., 2011*; *Liu et al., 2011*) by interacting with RIMs and $Ca^{2+}$ channels (*Wang et al., 2000*; *Hibino et al., 2002*). In addition, RIM-BPs, like RIM1/2, recruit Munc-13 and accelerate synaptic vesicle priming (*Brockmann et al., 2020*). At murine central synapses, RIM-BPs-deficient synapses demonstrate overall a rather mild decrease in the amounts of transmitter release (*Acuna et al., 2015*; *Grauel et al., 2016*; *Luo et al., 2017*; *Krinner et al., 2017*; *Brockmann et al., 2019*; *Butola et al., 2021*). Most studies so far have been mainly focused on synapses with high release probability (phasic synapses) such as the calyx of Held and CA3-CA1 synapses. That said, at mossy fiber-CA3 synapses, KO of RIM-BPs has a strong influence on synaptic transmission (*Brockmann et al., 2019*). Notably, at mossy fiber-CA3 synapses, release probability is relatively low (tonic synapses). It was unknown whether the reduction of transmitter release at RIM-BP2 KO mossy fiber synapses was due to impaired $Ca^{2+}$ channel recruitment and/or impaired vesicle docking or priming. Using STED microscopy, *Brockmann et al., 2019* suggested that the physical recruitment of Munc13-1 might be an important function of RIM-BP2, though functional demonstration had been lacking.

By using direct presynaptic patch-clamp recordings, we here observed a decrease of $Ca^{2+}$ current amplitudes (~30%) in RIM-BP2 KO mice (*Figure 1*). Consistently, STED microscopy supported reduced abundance of P/Q-type $Ca^{2+}$ channels (Cav2.1) in the mutant mossy fiber terminal (*Figure 7*). Interestingly, this observation is similar to that at *Drosophila* NMJ and hair cell synapses (*Liu et al., 2011*; *Krinner et al., 2017*), but not that at other synapses (*Acuna et al., 2015*; *Grauel et al., 2016*; *Butola et al., 2021*), suggesting that the functional role of RIM-BP2 in recruiting $Ca^{2+}$ channels differs among synapse types.

*Brockmann et al., 2019* observed a reduction of Munc13-1 cluster number and docked vesicles in RIM-BP2-deficient synapses at hippocampal mossy fiber terminals (*Figure 7*). They identified the terminals by using ZnT3, enriched at mossy fiber (*Brockmann et al., 2019*; *Wenzel et al., 1997*). Our STED imaging did not detect differences in Munc13-1 cluster number at the AZ between WT and RIM-BP2 KO mice. Notably, we analyzed the number of Cav2.1, Cav2.2, and Munc13-1 clusters only within AZs showing direct co-localization with VGLUT1 and PSD-95, which mainly reflects the AZs facing CA3 pyramidal cells rather than the synapses onto interneurons (*Acsády et al., 1998*; *Rollenhagen et al., 2007*). Such a difference in areas analyzed between previous and our study might explain the difference. In addition, some developmental variability/compensation could still occur in the range we studied and be relevant for the discrepancy between Munc13-1 results between *Brockmann et al., 2019* and this study. Also, the spatial resolution of our study is limited and we cannot rigidly measure the signals only from large mossy fiber terminals. It is important to state that our results do not exclude roles of RIM-BP2 in synaptic vesicle priming. Indeed, we have shown that the reduction of the EPSC amplitudes could not be fully explained by the reduced $Ca^{2+}$ influx (*Figure 5*). Because the differences in the RRP size measured from capacitance measurements were relatively minor (*Figures 1–3*), the fusion competence was likely to be impaired by RIM-BP2 KO, but this could be overcome by large $Ca^{2+}$ influx. Consistently, the EPSCs responses of KO became larger and comparable to those of WT (*Figure 6A and B*, *Figure 6—figure supplement 1*) during repetitive AP stimulation, at least qualitatively.

Recent studies have reported that RIM-BPs deletion altered $Ca^{2+}$ channel localization and loosened the coupling between $Ca^{2+}$ channels and synaptic vesicles at mammalian synapses (*Acuna et al., 2015*; *Grauel et al., 2016*; *Butola et al., 2021*). We investigated the $Ca^{2+}$ channel–vesicle coupling at hippocampal mossy fiber terminals. The sensitivity of release to $Ca^{2+}$ chelator EGTA was not changed between WT and RIM-BP2 KO mice. We also indicated RIM-BP2 KO did not alter the coupling of $Ca^{2+}$ channels to release sites by using STED microscopy, though spatial resolution of STED microscopy in this study could not allow to detect small changes (*Figure 7*).

At some mammalian synapses, deletion of RIM-BP2 affects physical distance between $Ca^{2+}$ channels and synaptic vesicles (*Acuna et al., 2015*; *Grauel et al., 2016*; *Butola et al., 2021*). In contrast, at

hippocampal mossy fiber terminals having low release probability (*Vyleta and Jonas, 2014*), RIM-BP2 KO decreased the abundance of P/Q-type $Ca^{2+}$ channels, thereby reducing $Ca^{2+}$ influx and neurotransmitter release. We hypothesize the specific molecular–architectural and biochemical contributions by RIM-BPs might be of different relevance among different types of synapses. *Brockmann et al., 2020* proposed that RIM-BP2 controlled $Ca^{2+}$ channel recruitment and vesicle priming by profound interaction with RIMs and Munc13s. These results have been obtained by studying hippocampal cultures dominated by phasic synapses. It seems possible that the interaction among three protein types is tight at some types of phasic synapses, simply because of the high density of these proteins at AZs. Alternatively, one might speculate about additional proteins, different interaction surfaces taken, and transsynaptic columns (*Tang et al., 2016*). Deletion of RIM-BP2 does not lead to loss of $Ca^{2+}$ channels themselves and/or vesicle priming due to remaining RIMs, Munc13, and other proteins, which can regulate recruitment of $Ca^{2+}$ channels and synaptic vesicle priming without RIM-BP2. Instead, fine-tuning of release such as synchronization is compromised. We hypothesize that AZ scaffold might be differently built, leading to different consequences when eliminating one component. Hence deletion of RIM-BP2 leads to immediate loss of $Ca^{2+}$ channels and vesicle priming. This is in line with the proposal that at hippocampal mossy fiber synapses, synaptic vesicles are only loosely coupled with $Ca^{2+}$ channels (*Vyleta and Jonas, 2014*) and vesicles are de-primed under resting conditions (*Miki et al., 2016*; *Neher and Brose, 2018*). Future research along this line may lead to an understanding of how fine molecular differences in the AZ scaffolds might orchestrate synaptic diversity.

Although this study reveals reduction of $Ca^{2+}$ currents and reduced fusion competence in RIM-BP2 KO mossy fiber synapses, it has some limitation. Most importantly, with conventional knockouts the observed changes could reflect compensatory developmental changes and not direct outcomes of the RIM-BP2 knockout. This issue should be resolved by using region-specific and time-dependent conditional KO.

## Materials and methods
### Slice preparation
All animal experiments were conducted in accordance with the guidelines of the Physiological Society of Japan and were approved by Doshisha University Animal Experiment Committee (A22063, D22063).

We used male and female C57BL/6 mice (postnatal days 35–40). WT and RIM-BP2 (encoded by the *Rimbp2* locus) KO mice (*Grauel et al., 2016*) were deeply anesthetized with isoflurane and decapitated. Their brains were quickly removed and chilled in sherbet-like cutting solution containing (in mM) 87 NaCl, 75 sucrose, 25 $NaHCO_3$, 1.25 $NaH_2PO_4$, 2.5 KCl, 10 glucose, 0.5 $CaCl_2$, and 7 $MgCl_2$ bubbled with 95% $O_2$ and 5% $CO_2$ (*Hallermann et al., 2003*). Hippocampal slices (300 µm thick) were prepared from brains using a vibratome (VT1200S, Leica) in ice-cold cutting solution. After slicing, slices were incubated in cutting solution at 37°C for 30 min and subsequently kept at room temperature (22–25°C) up to 4 h.

### Whole-cell recordings
Electrophysiological recordings were performed in a recording chamber filled with the extracellular solution containing (in mM) 125 NaCl, 2.5 KCl, 25 glucose, 25 $NaHCO_3$, 1.25 $NaH_2PO_4$, 0.4 ascorbic acid, 3 myo-inositol, 2 Na-pyruvate, 2 $CaCl_2$, and 1 $MgCl_2$ saturated with 95% $O_2$ and 5% $CO_2$. In some experiments, the concentration of $CaCl_2$ was changed to 1 mM, 1.5 mM, and 4 mM. For presynaptic membrane capacitance measurements, 1 µM tetrodotoxin (TTX, Wako) was added to block $Na^+$ channels. Moreover, for recording $Ca^{2+}$ currents (*Figure 1B*), 10 mM TEA-Cl was added to block $K^+$ channels. Slices were visualized by an upright microscope (BX-51, Olympus). Whole-cell patch-clamp recordings were applied to hippocampal mossy fiber terminals and CA3 pyramidal cells at room temperature. The patch pipettes were filled with the intracellular solution containing (in mM) 135 Cs-gluconate, 20 TEA-Cl, 10 HEPES, 5 $Na_2$-phosphocreatine, 4 MgATP, 0.3 GTP, and 0.5 EGTA (pH = 7.2 adjusted with CsOH). In some experiments, the concentration of EGTA was changed to 5 mM. Presynaptic patch pipettes (BF150-86-10, Sutter Instrument) had a resistance of 15–20 MΩ and series resistance ($R_s$) was 30–70 MΩ. $R_s$ was compensated so that residual resistance was 30–35 MΩ. For postsynaptic recordings, the same internal solution was used, except that EGTA concentration

was raised to 5 mM. Postsynaptic patch pipettes (BF150-86-10, Sutter Instrument) had a resistance of 3–7 MΩ and series resistance ($R_s$) was 5–20 MΩ, which was compensated by 50–80%.

Patch-clamp recordings were performed using an EPC10/3 or EPC10/2 amplifier (HEKA) in voltage-clamp mode, controlled by Patchmaster software (HEKA). Membrane currents were low-pass filtered at 2.9 kHz and sampled at 20 or 50 kHz.

Membrane capacitance measurements were performed using an EPC10/3 amplifier in the sine + DC configuration (*Lindau and Neher, 1988*; *Gillis, 2000*) using Patchmaster software (HEKA). A sine wave (30 mV in amplitude, 1000 Hz in frequency) was superimposed on the holding potential of –80 mV. The terminal was depolarized from –80 mV to +10 mV for 5–100 ms. The IV of presynaptic Ca currents were measured by sequential depolarized for 5 ms, with 2 s intervals from –80 mV to +70 mV by 10 mV steps.

For the EPSC measurements, postsynaptic cells (CA3 pyramidal cells) were voltage clamped at –60 to –80 mV. For *Figures 5 and 6*, the data were corrected for the different holding potential (re-calculated to the value at –70 mV assuming the reversal potential of 0 mV, which may have the correction effect of ~15%). At the end of the experiments, DCG-IV was applied to confirm mossy fiber synaptic responses, whenever possible. The mossy fibers were stimulated extracellularly using a glass electrode used for patch clamp containing the extracellular solution and the stimulation intensity was usually 10–60 V.

Data were obtained from at least three different WT and RIM-BP2 KO terminals in each set of experiments (biological replicate).

## Immunostaining

Brains of WT and RIM-BP2 KO mice were quickly removed and transferred to cryomold (Sakura Finetek) filled with tissue freezing medium (Leica). Brains were instantaneously frozen on aluminum block in liquid nitrogen and stored at –80°C. For cryosectioning, frozen brains were kept in cryostat (CM1860, Leica) at –18°C for 30 min. Then, 10-μm-thick sections were sliced and collected on cover glasses (Matsunami glass, 25 × 25 No.1). Sections were quickly fixed by dehydration with a heat blower at 50°C for 1 min and further dehydrated in ethanol for 30 min at –25°C and in acetone for 10 min on ice. After blocking with PBS containing 0.3% BSA at room temperature, sections were incubated with primary antibodies diluted in PBS containing 0.3% BSA: anti-VGLUT1 (1:2000, Synaptic Systems, guinea pig polyclonal, RRID:AB_887878), anti-PSD-95 (1:50, NeuroMab, mouse monoclonal IgG2a, clone K28/43, TC Supernatant, RRID:AB_2877189), anti-Munc13-1 (1:1000, mouse monoclonal IgG1, clone 11B-10G, *Sakamoto et al., 2018*), and anti-Cav2.1 (1:400, Synaptic Systems, rabbit polyclonal, RRID:AB_2619841) or anti-Cav2.2 (1:400, Synaptic Systems, mouse monoclonal IgG2b, clone 163E3, RRID:AB_2619843) or anti-RIM-BP2 (1:200, mouse monoclonal IgG2b, clone 8-4G, *Sakamoto et al., 2022*) for 3 hr at room temperature. Afterward, sections were washed and incubated with fluorescence dye-labeled secondary antibodies: DyLight 405 Anti-Guinea Pig IgG (Jackson ImmunoResearch, RRID:AB_2340470), Alexa Fluor 488 (Thermo Fisher Scientific)-labeled Anti-Mouse IgG2a (Jackson ImmunoResearch, RRID:AB_2338462), STAR635P (Abberior)-labeled Anti-Mouse IgG1 (Jackson ImmunoResearch, RRID:AB_2338461), and Alexa Fluor 594 (Thermo Fisher Scientific)-labeled Anti-Rabbit IgG (Jackson ImmunoResearch, RRID:AB_2340585) or Anti-Mouse IgG2b (Jackson ImmunoResearch, RRID:AB_2338463) for 1 hr at room temperature. Sections were post-fixed with 4% PFA in PBS and washed with PBS. Sections were mounted on coverslips using Prolong Glass Antifade Mountant (Thermo Fisher Scientific). The same number of sections were obtained from WT and RIM-BP2 KO brains and processed in parallel at the same experimental day (biological replicate).

For *Figure 7—figure supplement 3*, only WT brains were used. Primary antibodies used were anti-VGLUT1 (1:2000, Synaptic Systems, rabbit polyclonal, RRID:AB_887877), anti-VGLUT1 (1:2000, Synaptic Systems, guinea pig polyclonal, RRID:AB_887878), anti-mGluR1α (1:200, Nittobo Medical, guinea pig polyclonal, mGluR1a-GP-Af660), anti-GluA4C (1:100, Nittobo Medical, rabbit polyclonal, GluR4C-Rb-Af160), and anti-Munc13-1 (1:1000, mouse monoclonal IgG1, clone 11B-10G).

## STED imaging

STED imaging was performed using TCS SP8 STED 3× microscope (Leica) equipped with a 405 nm diode laser, a pulsed white light laser (WLL), a continuous 592 nm STED laser for alignment, a pulsed 775 nm STED laser, HyD detectors, and a 100× oil-immersion objective lens (NA = 1.4). STED images

were acquired using the Leica LAS-X software with an image format of 1024 × 1024 pixels, 16-bit sampling, 8-line accumulations, and 11.36-zoom factor, yielding a pixel size of 10 nm. HyD detectors were configured to counting mode with a gating from 0.5 to 6.5 ns. A 405 nm diode laser was used to excite DyLight405. Alexa488, Alexa594, and STAR635P were excited using WLL at 488 nm, 561 nm, and 633 nm, respectively. The 775 nm STED laser power was set to 75 and 100% of maximum power for depletion of STAR635P and Alexa594, respectively, and delay time was set to 300 ps. STED imaging was performed on thin sections obtained from four different WT and RIM-BP2 KO mice (biological replicate). Images were acquired three times from each section (technical replicate).

## Analysis

For electrophysiological experiments, the data were analyzed using Igor Pro (WaveMetrics) or Excel (Microsoft Corp). Presynaptic calcium currents were measured at the peak amplitude during the depolarizing pulse. Capacitance jumps were measured between baseline (before the pulse) and ~10 ms after the pulse where the effect of tail currents became minor. To determine the time constant of release, an exponential fit was applied. Values are given as mean ± SEM, and n indicates the number of recorded terminals or the number of animals used. Statistical method of sample size determination was not done, but our sample sizes are similar to those of previous studies (*Hallermann et al., 2003*; *Midorikawa and Sakaba, 2017*).

For STED imaging, data were analyzed using custom-designed programs in Mathematica (Wolfram), values were processed using Igor Pro (WaveMetrics), and pictures were created with ImageJ (NIH). To determine the area of AZs, image masks were generated from STED images of Munc13-1 by unsharp masking and image binarization. Only AZs that show co-localization with VGLUT1 and PSD-95 immunofluorescence signals were included in the analysis. To confine the analysis to the large mossy fiber boutons, AZs in presynaptic terminals of small VGLUT1-positive area were excluded. Then, the background-subtracted integral of signal intensity of Cav2.1 or Cav2.2 was quantified using the image masks of AZs. The background signal intensity was estimated from non-AZ area in the same STED images. For quantification of Cav2.1 clusters in the AZs, STED images of Cav2.1 were deconvolved using a Gaussian kernel with a radius of 40 nm. Image masks of each $Ca^{2+}$ channel were generated from deconvolved STED images by unsharp masking and image binarization. Then, the total signal intensity, the size, and the number of Cav2.1 clusters at the AZs were quantified using the image masks of Cav2.1 clusters. Custom-designed programs in Mathematica (Wolfram) are available as *Figure 7—source code 1*. Values are given as mean ± SEM, and n indicates the number of animals. Statistical method of sample size determination was not done, but our sample sizes are similar to those of previous studies (*Brockmann et al., 2019*).

Experiments were not fully randomized and blinded.

Statistical analysis was done in MATLAB (The MathWorks) or SPSS (IBM). *t*-test, ANOVA, and mixed model were used for statistical tests. Significance level was set at $\alpha = 0.05$, and p values are described in the main text or figure legends.

## Acknowledgements

We thank Stephan Sigrist, Dietmar Schmitz, and Christian Rosenmund for kindly providing us RIM-BP2 KO mice. We also thank Stephan Sigrist for comments on the manuscript and The IRCN Imaging Core, The University of Tokyo Institutes for Advanced Studies, for the use of STED microscopy and for assistance. This work has been supported by JSPS KAKENHI (JP20J20550 to RM, JP20KK0171, JP21K15183 to HS, JP20H03427 to KH, JP20KK0171, JP21H02598 to TS), JSPS Core-to-Core program A. Advanced Research Networks (JPJSCCA20220007 to TS), JST PRESTO (JPMJPR21E7 to HS), and Takeda Science Foundation (Bioscience and Specific Research Grants to TS).

# Additional information

## Funding

| Funder | Grant reference number | Author |
| --- | --- | --- |
| Japan Society for the Promotion of Science | KAKENHI JP20J20550 | Rinako Miyano |
| Japan Society for the Promotion of Science | JSPS Core-to-Core Program A. Advanced Research Networks JPJSCCA20220007 | Takeshi Sakaba |
| Japan Science and Technology Agency | PRESTO JPMJPR21E7 | Hirokazu Sakamoto |
| Takeda Science Foundation | Bioscience and Specific Research Grants | Takeshi Sakaba |
| Japan Society for the Promotion of Science | KAKENHI JP20KK0171 | Takeshi Sakaba Hirokazu Sakamoto |
| Japan Society for the Promotion of Science | KAKENHI JP21K15183 | Hirokazu Sakamoto |
| Japan Society for the Promotion of Science | KAKENHI JP20H03427 | Kenzo Hirose |
| Japan Society for the Promotion of Science | KAKENHI JP21H02598 | Takeshi Sakaba |
| Japan Society for the Promotion of Science | KAKENHI JP22K18358 | Kenzo Hirose |

The funders had no role in study design, data collection and interpretation, or the decision to submit the work for publication.

## Author contributions

Rinako Miyano, Conceptualization, Data curation, Formal analysis, Validation, Investigation, Writing - original draft, Writing - review and editing; Hirokazu Sakamoto, Conceptualization, Resources, Formal analysis, Funding acquisition, Investigation, Writing - original draft, Writing - review and editing; Kenzo Hirose, Conceptualization, Supervision, Funding acquisition, Investigation, Writing - review and editing; Takeshi Sakaba, Conceptualization, Supervision, Funding acquisition, Validation, Investigation, Writing - review and editing

## Author ORCIDs

Rinako Miyano http://orcid.org/0000-0002-0022-7198
Hirokazu Sakamoto http://orcid.org/0000-0002-9881-5173
Kenzo Hirose http://orcid.org/0000-0002-8944-6513
Takeshi Sakaba http://orcid.org/0000-0003-0688-7717

## Ethics

All animal experiments were conducted in accordance with the guidelines of the 599 Physiological Society of Japan, and were approved by Doshisha University Animal 600 Experiment Committee (A22063, D22063).

Reviewer #1 (Public Review): https://doi.org/10.7554/eLife.90799.3.sa1
Reviewer #2 (Public Review): https://doi.org/10.7554/eLife.90799.3.sa2
Author Response https://doi.org/10.7554/eLife.90799.3.sa3

# Additional files

## Supplementary files
• MDAR checklist

## Data availability

All data generated or analysed during this study are included in the manuscript and supporting files. Numerical values of graphs are provided in source data files. The custom-written code files in Mathematica are uploaded as *Figure 7—source code 1*.

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
