## [Editor Report · eLife assessment]

Miyano et al. study the impact of RIM-BP2 deletion at mossy fiber synapses using direct electrophysiological recordings from mossy terminals and STED super-resolution microscopy. The article addresses an **important** question in the field of synaptic transmission and provides **compelling** evidence demonstrating reduced calcium channel abundance in mossy terminals upon RIM-BP2 removal.

---

## [Referee Report · Reviewer #1 (Public Review)]

This interesting study by Miyano combines slice electrophysiology and superresolution microscopy to address the role of RBP2 in Ca2+ channel clustering and neurotransmitter release at hippocampal mossy fiber terminals. While a number of studies demonstrated a critical role for RBPs in clustering Ca2+ channels at other synapses and some provided evidence for a role of the protein in molecular coupling of Ca2+ channels and release sites, the present study targets another key synapse that is an important model for presynaptic studies and offers access to a microdomain controlled synaptic vesicle (SV) release mechanism with low initial release probability.

Summarizing a large body of high quality work, the authors demonstrate reduced Ca2+ currents and a reduced release probability. They attribute the latter to the reduced Ca2+ influx and can restore release by increasing Ca2+ influx. Moreover they propose an altered fusion competence of the SVs, which is not so strongly supported by the data in my view.

The effects are relatively small, but I think the careful analysis of the RBP role at the mossy fiber synapse is an important contribution.

---

## [Referee Report · Reviewer #2 (Public Review)]

Summary: The proper expression and organization of CaV channels at the presynaptic release sites are subject to coordinative and redundant control of many active zone specific molecules including RIM-BPs. Previous studies have demonstrated that ablation of RIM-BPs in various mammalian synapses causes significant impairment of synaptic transmission, either by reducing CaV expression or decoupling CaV from synaptic vesicles. The mechanisms remain unknown.

In the manuscript, Sakaba and colleagues aimed to examine the specific role of RIM-BP2 at the hippocampal mossy fiber-CA3 pyramidal cell synapse, which is well-characterized by low initial release probability and strong facilitation during repetitive stimulation. By directly recording Ca2+ currents and capacitance jumps from the MF boutons, which is very challenging but feasible, they showed that depolarization-evoked Ca2+ influx was reduced significantly (~39%) by KO of RIM-BP2, but no impacts on Ca-induced exocytosis and RRP (measured by capacitance change). They used STED microscopy to image the spatial distribution of CaV2.1 cluster but found no change in the cluster number with slight decrease in cluster intensity (~20%). They concluded that RIM-BP2 function in tonic synapses by reducing CaV expression and thus differentially from phasic synpases by decoupling CaV-SV.

In general, they provide solid data showing that RIM-BP2 KO reduces Ca influx at MF-CA3 synapse, but the phenotype is not new as Moser and colleagues have also used presynaptic recording and capacitance measurement and shown that RIM-BP2 KO reduces Ca2+ influx at hair cell active zone (Krinner et al., 2017), although at different synapse model expressing CaV1.3 instead of CaV2.1. Further, the concept that RIM-BP2 plays diverse functions in transmitter release at different central synapses has also been proposed with solid evidence (Brockmann et al., 2019).

---

## [Author Response]

The following is the authors’ response to the original reviews.

**Public Reviews:**

**Reviewer #1 (Public Review):**
This nice study by Miyano combines slice electrophysiology and superresolution microscopy to address the role of RBP2 in Ca2+ channel clustering and neurotransmitter release at hippocampal mossy fiber terminals. While a number of studies demonstrated a critical role for RBPs in clustering Ca2+ channels at other synapses and some provided evidence for a role of the protein in molecular coupling of Ca2+ channels and release sites, the present study targets another key synapse that is an important model for presynaptic studies and offers access to a microdomain controlled synaptic vesicle (SV) release mechanism with low initial release probability.Summarizing a large body of high-quality work, the authors demonstrate reduced Ca2+ currents and a reduced release probability. They attribute the latter to the reduced Ca2+ influx and can restore release by increasing Ca2+ influx. Moreover, they propose an altered fusion competence of the SVs, which is not so strongly supported by the data in my view.The effects are relatively small, but I think the careful analysis of the RBP role at the mossy fiber synapse is an important contribution.

We thank the reviewer for careful assessment of the paper. We agree that while reduced Ca influx in KO is relatively straightforward, impaired priming is somewhat indirect, remaining as suggestion. We also noted that Moser and colleagues have analyzed the function of RIM-BP2 at hair cell synapses and also showed reduced Ca influx. In cortical synapses, there have been no study using direct presynaptic recording. In the revision, we carefully cited previous studies and tried to be fair. We hope that the current revision is much improved.

**Reviewer #2 (Public Review):**
The proper expression and organization of CaV channels at the presynaptic release sites are subject to coordinative and redundant control of many active zone-specific molecules including RIM-BPs. Previous studies have demonstrated that ablation of RIM-BPs in various mammalian synapses causes significant impairment of synaptic transmission, either by reducing CaV expression or decoupling CaV from synaptic vesicles. The mechanisms remain unknown.In the manuscript, Sakaba and colleagues aimed to examine the specific role of RIM-BP2 at the hippocampal mossy fiber-CA3 pyramidal cell synapse, which is well-characterized by low initial release probability and strong facilitation during repetitive stimulation. By directly recording Ca2+ currents and capacitance jumps from the MF boutons, which is very challenging but feasible, they showed that depolarization-evoked Ca2+ influx was reduced significantly (~39%) by KO of RIM-BP2, but no impacts on Ca-induced exocytosis and RRP (measured by capacitance change). They used STED microscopy to image the spatial distribution of the CaV2.1 cluster but found no change in the cluster number with a slight decrease in cluster intensity (~20%). They concluded that RIM-BP2 functions in tonic synapses by reducing CaV expression and thus differentially from phasic synapses by decoupling CaV-SV.In general, they provide solid data showing that RIM-BP2 KO reduces Ca influx at MF-CA3 synapse, but the phenotype is not new as Moser and colleagues have also used presynaptic recording and capacitance measurement and shown that RIM-BP2 KO reduces Ca2+ influx at hair cell active zone (Krinner et al., 2017), although at different synapse model expressing CaV1.3 instead of CaV2.1. Further, the concept that RIM-BP2 plays diverse functions in transmitter release at different central synapses has also been proposed with solid evidence (Brockmann et al., 2019).

We thank the reviewer for careful reading of the ms. We agree that previous studies have sown reduced Ca influx at hair cells, and diverse function of RIM-BP2 in different central synapses have been proposed by Brockman et al. The new point of this study is we firmly and quantitatively show the reduced Ca currents using direct presynaptic recording, which has not been done in mossy fiber synapses or cortical synapses in general. Quantitative and time-resolved measurements of the presynaptic currents cannot be done by other methods, so far. In this revision, we point this out carefully.

**Reviewer #1 (Recommendations For The Authors):**
The MS is overall carefully prepared and I have only a few minor comments to help with further improving the manuscript.Abstract:I think the notion of different RBP function at tonic and phasic synapses is not so well founded. The reduced number of Ca2+ channels and their altered topography have been shown in multiple synapses that also include those with phasic release. Quantitative structural and functional analysis of presynaptic Ca2+ channels of RBP-2 and RBP1-2 DKO deficient AZs closely related to the present study has e.g. been provided for auditory synapses e.g. hair cells, endbulb/calyx of end synapses that provide both phasic and sustained release.

In abstract, we have omitted description of phasic vs tonic synapses, because it is not well founded as the reviewer pointed out. Specifically, in abstract (Line 13~):

“Synaptic vesicles dock and fuse at the presynaptic active zone (AZ), the specialized site for transmitter release. AZ proteins play multiple roles such as recruitment of Ca2+ channels as well as synaptic vesicle docking, priming and fusion. However, the precise role of each AZ protein type remains unknown. In order to dissect the role of RIM-BP2 at mammalian cortical synapses having low release probability, we applied direct electrophysiological recording and super-resolution imaging to hippocampal mossy fiber terminals of RIM-BP2 KO mice. By using direct presynaptic recording, we found the reduced Ca2+ currents. The measurements of EPSCs and presynaptic capacitance suggested that the initial release probability was lowered because of the reduced Ca2+ influx and impaired fusion competence in RIM-BP2 KO. Nevertheless, larger Ca2+ influx restored release partially. Consistent with presynaptic recording, STED microscopy suggested less abundance of P/Q-type Ca2+ channels at AZs deficient in RIM-BP2. Our results suggest that the RIM-BP2 regulates both Ca2+ channel abundance and transmitter release at mossy fiber synapses.”

Intro:Line 48: consider adding Butola et al., 2021 /endbuld of Held to reference which concurs on the notion made for Calyx. However, a contrasting finding was made for another synapse with tight coupling: RBP2 deletion did not alter tight coupling in hair cells (Krinner et al., 2017). Line 51: RBP-DKO/lack of additional effect of RBP1 deletion: suggest adding Krinner et al., 2021 to reference, which concurs with the notion made for hair cells.

We cited Butola et al., 2021 (Line 49) and Krinner et al., 2021 (Line 52), as the reviewer suggested.

Results:STED microscopy: I am concerned with two aspects of the analysis/presentation. (I) I recommend replacing density with abundance as the authors do not resolve single channels. (II) I appreciate the note of caution about the fact that STED nanoscopy due to the non-linear nature of the depletion process should/could not be easily used to quantify copy numbers based on immunofluorescence. I would recommend the authors perform 2D Gaussian fitting to at least the Cav2.1 immunofluorescent spots neighboring Munc13-1 spots and report the short and long axis estimates as well as potentially the area. Should the authors have confocal Cav2.1 and Cav2.2 immunofluorescent data co-acquired with STED of Munc13-1, this would be very valuable additional information, but I do not think the experiment is essential for the sake of publication if it was not done already, given the large body of high-quality physiology data.

I) We have changed the term from density to abundance as the reviewer suggested throughout the manuscript.

II) As the reviewer suggested, we have carried out 2D Gaussian fitting of Cav2.1 spots. The length, width, and area of Cav2.1 clusters in the AZ were not different between WT and RIM-BP2 KO terminals (Line 431-433, Figure 7-figure supplement 4). The spatial resolution of STED, especially at mossy fiber synapses in the tissue, and a small difference between WT and KO (~30 % expected from electrophysiology) could prevent detection of the difference, unlike ribbon synapses and fly NMJ where release sites and Ca channel clusters are well defined. We should also note that the intensity was calculated similar to previous studies (integral of signal intensity, Krinner et al., 2017), and not absolute peak intensity.

As the reviewer suggested, we have added confocal data (Line 434-436, Figure 7-figure supplement 5). We have determined the AZ area from the Munc13-1 STED data, and Munc13-1, Cav2.1 and Cav2.2 intensities were quantified. As shown in the figure, only Ca2.1 intensity was reduced in KO, consistent with the STED data.

Nevertheless, we should be cautious about interpretation of the intensity as the reviewer suggested, and are aware that the data are just consistent with electrophysiology. From imaging, we only see a qualitative rather than quantitative difference between WT and KO.

Discussion:I think the focus on alterations of presynaptic Ca channels could be further strengthened along with the discussion of the relevant previous studies.

Thank you for the suggestion. We have added a paragraph as shown below in the discussion (Line 531~).

“By using direct presynaptic patch clamp recordings, we here observed a decrease of Ca2+ current amplitudes (~30%) in RIM-BP2 KO mice (Fig. 1). Consistently, STED microscopy supported reduced abundance of P/Q-type Ca2+ channels (Cav2.1) in the mutant mossy fiber terminal (Fig. 7). Interestingly, this observation is similar to that at *Drosophila* NMJ and hair cell synapses (Liu et al., 2011; Krinner et al., 2017), but not that at other synapses (Acuna et al., 2015; Grauel et al., 2016; Butola et al., 2021), suggesting that the functional role of RIM-BP2 in recruiting Ca2+ channels differs among synapse types. “

**Reviewer #2 (Recommendations For The Authors):**
Minor questions:1. The title is misleading as it only shows RIM-BP2 regulates CaV expression but not clustering.

This has been pointed out by the 1st reviewer, too. We have adopted the term “abundance” as suggested by the 1st reviewer and changed to “RIM-BP2 regulates Ca2+ channel abundance and neurotransmitter release at hippocampal mossy fiber terminals.”

2. Figure 7 legend. Again, RIM-BP2 only changes the intensity of CaV2.1 clusters but not the density.

Changed Figure 7 title from “RIM-BP2 deletion alters the density …” to “RIM-BP2 deletion alters the signal intensity …”.

3. Line 31: "Ca2+ influx through voltage-gated Ca2+ channels triggers neurotransmitter release from synaptic vesicles within a millisecond" is not correct. Ca-evoked transmitter release can only occur with such fast speed at very specialized synapses such as the calyx of Held but not at general chemical synapses.

We changed “within a millisecond” to “within milliseconds” (Line 30).

4. Line 44-46: In *Drosophila* NMJs and at *Drosophila* NMJs are redundant.

We eliminated “at *Drosophila* NMJs”.

5. The authors should use the verb tense consistently throughout the manuscript such as"In RIM-BP1,2 DKO mice, the coupling between Ca2+ channels and synaptic vesicles became loose, and action potential-evoked neurotransmitter release was reduced at the calyx of Held synapse (Acuna et al., 2015). At hippocampal CA3-CA1 synapses, RIM-BP2 deletion alters Ca2+ channel localization at the AZs without altering total Ca2+ influx. Besides, RIM-BP1,2 DKO has no additional effect...".

We changed verb tenses in Line 46-49, Line 55-58, and Line 62-67. We also checked the ms once more. Thank you for pointing this out.

6. Line 59: technically difficulty should be technical difficulty.

Fixed.

7. Figure 4A-B are representative traces of 0.5 mM EGTA (black) or 5 mM EGTA (red) recorded from the same terminals or from different terminals but simply superimposed?

Representative traces are recorded from different terminals. We describe this point in the figure legend (Fig 4A). We are very sorry for confusion.